# Fine Texture Detection Based on a Solid–Liquid Composite Flexible Tactile Sensor Array

**DOI:** 10.3390/mi13030440

**Published:** 2022-03-14

**Authors:** Weiting Liu, Guoshi Zhang, Binpeng Zhan, Liang Hu, Tao Liu

**Affiliations:** State Key Laboratory of Fluid Power and Mechatronic Systems, School of Mechanical Engineering, Zhejiang University, Hangzhou 310007, China; liuwt@zju.edu.cn (W.L.); gszhang@zju.edu.cn (G.Z.); 11825032@zju.edu.cn (B.Z.); cmeehuli@zju.edu.cn (L.H.)

**Keywords:** tactile sensor array, solid–liquid composite structure, stochastic resonance, texture detection, sensitivity, signal characteristic

## Abstract

Surface texture information plays an important role in the cognition and manipulation of an object. Vision and touch are the two main methods for extracting an object’s surface texture information. However, vision is often limited since the viewing angle is uncertain during manipulation. In this article, we propose a fine surface texture detection method based on a stochastic resonance algorithm through a novel solid–liquid composite flexible tactile sensor array. A thin flexible layer and solid–liquid composite conduction structure on the sensor effectively reduce the attenuation of the contact force and enhance the sensitivity of the sensor. A series of ridge texture samples with different heights (0.9, 4, 10 μm), different widths (0.3, 0.5, 0.7, 1 mm), but the same spatial period (2 mm) of ridges were used in the experiment. The experimental results prove that the stochastic resonance algorithm can significantly improve the signal characteristic of the output signal of the sensor. The sensor has the capability to detect fine ridge texture information. The mean relative error of the estimation for the spatial period was 1.085%, and the ridge width and ridge height, respectively, have a monotonic mapping relationship with the corresponding model output parameters. The sensing capability to sense a fine texture of tactile senor surpasses the limit of human fingers.

## 1. Introduction

The tactile sensor can provide useful contact information for robot feedback and control to better complete manipulation tasks in a complex environment. Existing research on tactile sensors has achieved monitoring of static contact information, such as temperature, curvature, and contact force [1,2,3], but the research on the acquisition of dynamic contact information is not thorough enough. Surface texture information, that is, the height, width, spatial period, and other texture shape information of the contacted object, is an important part of the dynamic contact information. It has been proved that the tactile sensing detection of the surface texture information of the object is helpful for dexterous manipulation [4,5] and minimally invasive surgery [6,7]. Vision is also an effective method to obtain surface texture information [8], but due to the uncertainty of the viewing angle during operation, the visual perception is often limited. In addition, it is noteworthy that the sensing process of the tactile sensor is the exploratory movement of the sensor sliding on the surface to obtain texture information, such as vibration information [9]. In other words, as long as the sensor has sufficient performance and can touch well the object, the surface texture information of the object can be obtained even under complex manipulation conditions.

Some existing researches use multi-array tactile sensors and fast Fourier transform (FFT, hereafter) technology to obtain characteristic signals and estimate the surface texture information of objects, which had certain limitations. In 2011, C.M. Oddo et al. [10,11] developed a 2 × 2 MEMS piezoresistive tactile sensor array with high spatial resolution and sensitivity and integrated it on the fingertips of humanoid robot fingers with bionic fingerprints. When actively touching, it could quantitatively evaluate the ridge texture with a spatial period difference as low as 40 μm (400, 440, 480 μm spatial periods, 1 mm ridge height). H.B. Muhammad et al. [12] proposed a capacitive tactile sensor array based on silicon MEMS, which resolved forces in the sub-mN range and successfully distinguished fine ridge textures with ridge heights of 200 μm and spatial periods as low as 200 μm. However, the research mentioned above on object surface information detection mainly focused on the estimation of the spatial period of ridge textures in the micron range and did not reach the capability to detect fine textures with ridge heights of several microns, such as human fingers [13,14]. In order to eliminate the influence of the difference between the mechanical properties of structural materials such as friction and frequency characteristics and human fingers on the sliding tactile signal, in 2019, Yasutoshi Takekawa et al. [15,16,17] developed a wearable skin vibration sensor to detect the skin-propagated vibration caused by finger rubbing an object surface, and successfully identified the spatial period and ridge height information of the fine ridge texture by calculating the power spectral density of the sensor output signal. The minimum ridge height of the fine texture exceeded the limitation to 25 μm, but in this situation, the characteristic of the signal was already very weak, and the detectable performance of the signal was limited.

Another commonly used method is using the tactile signal obtained by the designed tactile sensor for machine learning to classify and identify the surface texture of the object. In 2012, in order to achieve the ability of human fingertips to detect texture, Jeremy A. Fishel et al. [18,19] proposed Bio-Tac, a tactile finger with a solid–liquid composite structure, which could transmit the vibration of the skin through an incompressible liquid. The tactile finger could detect continuous vibrations as small as a few nanometers at 330 Hz and very small transient events that occurred when small particles fell on the finger, which was beyond human fingers’ capabilities. The Bayesian exploration method was used to classify 16 natural textures with an accuracy of 99.6%. In 2017, Udaya B. Rongala et al. [20,21] imported the output signal of the 2 × 2 piezoresistive tactile sensor array into the lzhikevich neuron model to enable neuromorphic artificial touch and simulate the firing pattern of mechanoreceptors in the skin. The generated neural spike pattern was used to perform neural network learning and classification of 10 natural textures, including glass materials with a grain size of less than 90 nm. In 2019, Sriramana Sankar et al. [5,22] designed a flexible bionic finger integrated with a 3 × 3 tactile sensor array. When the finger touched seven different texture samples (flat, ridges, bumps), the corresponding tactile response also transformed into a neural spike pattern with different characteristics that simulated the firing pattern of mechanoreceptors, and the support vector machine classifier was used to learn and classify with an accuracy of 99.21%. The above research revealed the capability of simulating human tactile behavior and accurately classifying different texture information, including fine textures, but the detailed texture information is still insufficient, and the complex structure encapsulation and machine learning algorithms limit their applications.

To sum up, research on texture detection needs more extensive and systematic investigations. FFT processing directly on the tactile signals can quickly and easily obtain texture feature information, such as the spatial period and ridge height of the ridge texture. However, due to the limited performance of most simple-structured tactile sensors to discern the tiny structural variation, FFT processing does not perform well in situations of fine texture detection. Thus, a tactile sensor with a higher sensitivity is needed. However, the capability of the existing simple tactile sensor to detect weak force changes is still far behind that of humans [23]. Another common processing method is the machine learning method. The tactile signals are used to train the models to accurately classify different textures, but that needs to go through a complicated learning process, and detailed feature information of the surface textures cannot be detected. Therefore, a universal, real-time method based on a tactile sensor with a simple structure for fine texture detection is extremely needed.

In the process of detecting fine textures, the signals generated by the tactile sensor are weak and may even be submerged in system noise, such as touch action noise, circuit noise, etc., resulting in its characteristics being difficult to detect. The stochastic resonance algorithm (SR, hereafter) is a commonly used weak signal detecting method, which adds random noise to the weak signal to improve the detectability of the target signal or enhance the accuracy of information interpretation [24]. Through a series of psychophysical experiments, Kadir Beceren et al. [25,26] proved that SR could affect the difference threshold of human tactile sensation, and appropriate noise can enhance the accuracy of tactile sensing. Yuichi Kurita et al. [27] used the SR to develop surgical forceps with sensorimotor-enhancing capabilities and verified its texture detection capability, which was of great help to surgeons in laparoscopic surgery. To this end, we have introduced the SR into our tactile sensor system to enhance the capability of our tactile sensor for fine textures detection.

In this article, we use a 2 × 2 solid–liquid composite flexible tactile sensor array to obtain tactile information. A method of fine texture detection based on SR is put forward. By adding random noise to the output signal of the tactile sensor and inputting the mixed signal to a bistable SR model, we obtain tactile signals with strong signal characteristics and figure out the corresponding texture information through numerical analysis. The presence of the flexible layer will lead to a decrease in the transmission efficiency of the contact force, and the thicker the flexible layer, the greater the attenuation of the contact force [28,29]. It is meaningful to use a solid–liquid composite conduction structure to reduce the attenuation of the contact force through a flexible layer. Most of the existing tactile texture detection methods are aimed at the detection of micron-level fine texture or can only classify a limited number of textures. The method proposed in this article has the potential to detect sub-micron fine ridge texture. In addition, the authors estimate the texture features of the fine ridge texture, such as the spatial period, ridge height, and ridge width. At last, the authors present the signal characteristic change of the output tactile signal and the error of the texture characteristic value.

The rest of this article is organized as follows. The tactile sensor device is presented in Section 2, followed by the introduction of the tactile signal processing methodology based on SR in Section 3. Detailed experiments and experimental results are presented in Section 4. Subsequently, the results are commented on in Section 5. Finally, the conclusion and prospect are discussed in Section 6.

## 2. Tactile Senor Device

In this article, we propose a 2 × 2 piezoresistive solid–liquid composite flexible sensor array to detect fine texture based on previous research in the laboratory [30]. Piezoresistive sensors are small in size, have a good frequency response, and are especially sensitive to small pressure differences, which are easier to meet our requirements for tactile sensor arrays. The structure of the sensor is a thin flexible layer encapsulating the fluid and the silicon-sensing elements inside. We replaced some parts of the traditional single flexible layer with incompressible liquids to utilize the characteristics of high hydraulic transmission efficiency to reduce the attenuation of the contact force. Different from traditional hydraulic devices, the fluid chamber of this sensor cell is made of soft material, and there is a coupling effect between its deformation and the encapsulated liquid compression. Therefore, when the flexible layer is deformed by external pressure and compresses the liquid inside, the encapsulated liquid will generate pressure to act on the sensing elements, causing it to produce a corresponding output. After the output signal is conditioned by the NSA2860 chip, the encapsulated sensing cell has good sensing performances, with a sensitivity of 3.46 mV/mN, linearity of 0.996, repeatability error of 1.2%, and hysteresis error of 1.5%. The force sensing range is from 1 mN to 1.5 N.

The structure of the sensing cell is shown in Figure 1a. The sensor array mainly consists of four parts: flexible printed circuit board (FPCB, hereafter) base, silicon pressure sensing die, silicone oil liquid layer, and polydimethylsiloxane (PDMS, hereafter) layer. The overall size is 12 mm × 12 mm × 2.5 mm, including four sensor cells in a 2 × 2 layout, with the spacing of sensor cells of 5.6 mm. The thickness of the PDMS flexible layer is 0.6 mm. The selected piezoresistive sensor element was a cubic silicon pressure sensor die (Silicon Microstructures, Inc., SM5108C-060, Milpitas, CA, USA). The silicone oil liquid layer was filled with 50cSt silicone oil (Dow Corning Inc. pmx-200, Midland, MI, USA). The silicone oil liquid not only acts as a transferring medium of pressure conduction but also effectively protects the silicon-based sensing element. The schematic diagram of the sensor array is shown in Figure 1b. To sum up, the whole sensor has a size similar to a human fingertip and is made of flexible material, which is easy to integrate on the prosthetic fingertip with various curves on its surface.

The sensor fabrication process is shown in Figure 1c. Firstly, paste the sensing element on the FPCB base and connect the four pads on the sensing element and the corresponding pads on the FPCB base with gold wires, respectively; secondly, design the mold and obtain the shaped PDMS layer through casting and curing, then paste it on the FPCB base to form the liquid cavity; thirdly, evacuate the air in the liquid cavity and fill it with silicone oil in the vacuum box; fourthly, take the sensor out and seal the structure with sealant; finally, integrate the prepared sensor array on the fingertip of the prosthesis.

## 3. Methodology

### 3.1. Texture Detection

The following is the sensing method for texture detection based on the designed tactile sensor. As shown in Figure 2, the feature information required to describe the ridge texture of a sample includes the spatial period *sp*, the ridge height *rh*, and the ridge width *rw*. The tactile sensor array with sensing cell spaced *d* is integrated on the prosthetic fingertip and slides horizontally across the surface of the texture sample at a certain speed *v* from left to right. The two adjacent sensing cells successively contact the ridge of the sample and are both affected by normal force *F* so that the sensing element inside them generate corresponding voltage signals. Since the ridges are uniformly distributed on the sample surface, these signals have a period of *T*; in other words, they have a principal frequency of *f_prin_*. Moreover, the two adjacent sensing cells will successively generate the same signal for the same ridge. The time interval between the two signals is ∆*t*. It can be easily figured out that the spatial period *sp* and the ridge width *rw* are related to the characteristic parameter *f_prin_* × ∆*t*, which is(1)$$\mathrm{sp}=\frac{d}{f_{\mathrm{prin}\;\times \;∆t}},\;\mathrm{rw}=n\;\times \;\mathrm{sp}$$
where *n* is the ratio of the ridge width to the spatial period in the ridge texture, which can be obtained from the time domain information of the signal. The ridge height *rh* is related to the amplitude information *A* of the principal frequency of the signal [30].

This article focuses on the detection capability of the tactile sensor array for fine ridge textures whose ridge height *rh* less than 10 μm (this height is close to the detection limit of a human finger). The signal generated when the tactile sensor array slides over the ridge texture surface is mainly composed of two-state voltage signals, which correspond to the positions of the ridges and grooves, respectively. However, because the height difference between ridges and grooves is small and the detection capability of the tactile sensor array is limited by its sensitivity, the difference between the two signal states is not obvious, and the signals may even be submerged by system noise. Therefore, this article mainly discussed a method to enhance signal characteristics of weak bistable signals to improve the detection capability of the tactile sensor array.

### 3.2. Stochastic Resonance Method

The bistable system stochastic resonance method is chosen to enhance the signal characteristic of the tactile signals. When the tactile sensor array slides over the surface of the object, it will sense the surface texture information of the object and transmit signals in time through a limited number of sensing cells. Especially in the process of fine texture detection, the generated tactile signals have two states, and its characteristic is weak. The bistable system stochastic resonance method can effectively enhance some deterministic characteristics of weak tactile signals, which is suitable for the investigation of fine texture information detection.

As mentioned in Section 3.1, the experimental samples in this article are fine textures with ridge heights less than 10 μm, which makes the signal characteristic of the two-state signals generated by the tactile sensor weak. When the prepared tactile sensor slides across the ridge texture with a right height *rh* of 0.9 μm at a certain speed *v*, the time-domain signal generated has a weak periodicity, as shown in Figure 3a. After FFT processing, its frequency-domain signal has a prominent amplitude at its principal frequency, as shown in Figure 3b. However, it is difficult to obtain detailed texture information such as ridge width due to the excessive system noise mixed in the time-domain signal. Therefore, we first established a suitable bistable system stochastic resonance model to enhance the signal characteristic of weak tactile signals. Then, the output signal is processed by FFT to obtain relevant information, such as the principal frequency and the amplitude of the principal frequency, so as to obtain more feature information of the fine ridge texture.

In this article, a bistable system stochastic resonance model based on the Langevin equation is established, which is
(2)$$\{\begin{array}{c} \dot{x\;}{=-V}^{’}\left(x\right)\;+\;K(s\left(t\right)\;+\;\Gamma \left(t)\right)\\ <\Gamma \left(t\right)>=0,\;<\Gamma \left(t\right),\Gamma \left(0\right)\;>\;=2D\delta (t) \end{array}$$
where *V*(*x*) is the nonlinear bistable potential function, and
(3)$$V\;(x)=-\frac{a}{2}x^{2}+\frac{b}{4}x^{4}\;a\;>\;0,\;b\;>\;0$$
*s*(*t*) is the periodic tactile signal; *Γ*(*t*) is the input Gaussian white noise with intensity *D* and 0 mean value. *K* is the normalized transformation factor of the input mixed signal. By comparing the frequency range of the generated tactile signal and the intensity ($\sqrt{D}$ = 0.3 mV) of input Gaussian white noise with standard system parameters, we determine the system parameter (*a* = *b* = 16, *K* = 130,000). The determined bistable system is shown in Figure 4. The system will appear stochastic resonance effect under the common drive of the input weak periodic tactile signal and Gaussian white noise so that the system state jumps between the two steady states. The output is a bistable signal with characteristic information of the original signal.

## 4. Experiment

### 4.1. Experiment Platform

The experimental loading platform for the object surface texture detection experiment is shown in Figure 5. The prosthetic fingertip integrated with the tactile sensor array was mounted on a motion platform under position/velocity control. The motion platform is composed of a three-axis precision linear platform (M-VP-25XL-XYZ, Newport Corporation, Irvine, CA, USA) having travel of 25 mm along each axis and a long-travel linear motor stage (M-IMS600LM, Newport Corporation, Irvine, CA, USA) having a maximum speed of 500 mm/s with a minimum incremental motion of 20 nm. Thus, the prosthetic fingertips on the motion platform can touch the texture samples with different forces and scan across the surface texture sample with various velocities. The ridge texture sample was mounted on a fixed stage by a designed clamp. The output voltage generated by the piezoresistive die in the tactile sensor array is conditioned by the NSA2860 chip and then are collected by a NI-DAQ device (USB-6212, National Instruments Corporation, Austen, TX, USA). The ridge texture samples are a series of silicon wafers with different ridge heights and ridge widths, which are produced by exposure, development, and etching using a photolithography method, as shown in Figure 5b,c. A total of 12 texture samples are presented, and detailed parameters of them are shown in Table 1.

### 4.2. Experiment Detail

The main experimental process is as follows. As shown in Figure 6, the sensing cells are distributed on the surface of the tactile sensor in a 2 × 2 array, numbered 1#, 2#, 3#, 4#, respectively, and the distance of two adjacent sensing cells along the *x*-axis and *y*-axis are equal. When the tactile sensor starts from the initial position and slides across the sample surface along the *x*-axis direction that is perpendicular to the ridge texture at a certain speed, we can obtain the time interval ∆*t* of the signals generated by adjacent sensing cells for the same ridge. The positive *x*-axis is selected previously as the sliding direction of the tactile sensor; thus, the signals generated by the sensing cells 1#, 3# and 2#, 4# form a signal group with time-domain correlation, respectively and the two signal groups are in a control relationship with each other. Selected the signal group (1#, 3#) as the experimental result to validate the method proposed in this article.

The processing flow of the raw output signal of sensing cell 3# is shown in Figure 7. The sample used is the S31 sample with a ridge height of 0.9 μm, a ridge width of 300 μm, and a spatial period of 2000 μm. Due to the inevitable irregular jitter of the linear stage during the sliding experiment, the signal exhibited a baseline drift phenomenon, as shown in Figure 7a. The signal passed through a high-pass filter with a cut-off frequency of 1 Hz to remove low-frequency interference, and we obtained the original signal with a baseline at 0, as shown in Figure 7b. After inputting the original signal and the Gaussian white noise with a certain noise intensity into the bistable system stochastic resonance model based on the Langevin equation, the first-order ordinary differential Equations (2) and (3) are solved by the fourth-order Runge–Kutta algorithm. The proper noise intensity of Gaussian white noise can make the output signals have a larger signal-to-noise ratio (SNR, hereafter), but because the amplitude difference of the input signals is small and the optimal SNR of output is not the goal, we selected the same noise intensity through prior experiments and generated the noise signal with the Matlab function. The signal processed by SR is shown in Figure 7c, and its signal characteristic has been significantly enhanced. Since the signals have obvious characteristics, the time interval Δ*t* can be easily obtained through correlation analysis with the processing result of the signal generated by sensing cell 1#. The frequency-domain signal obtained by FFT processing contains the principal frequency information *f_prin_* and its amplitude information *A*. The difference between the two states of the time-domain signal is obvious, but there is still a certain amount of noise that interferes with the acquisition of texture information. Passed the randomly intercepted signal segment through a software-designed threshold comparator with a threshold of 1 V to obtain a state-unified signal, as shown in Figure 7d. The ratio of the high-level state of the signal segment to the level period is calculated and mapped to the ratio *n* of the ridge width to the spatial period in the ridge texture. The texture feature information of the corresponding sample can be figured out by combining the parameter results obtained above.

### 4.3. Experiment Result

Signals processed by SR exhibit a stronger signal characteristic in both time and frequency domains. Figure 8 shows the output voltages (i.e., 1# and 3#) of two piezoresistive die located at two adjacent tactile sensing cells. These signals have been previously processed in advance with a high-pass filter to remove low-frequency interference. As shown in Figure 8a, it is observed that the output original signal has strong random noise, especially, there is only slight periodic vibration when the tactile fingertip slides over the fine texture sample. Meanwhile, the time interval, as described in Section 3.1, is difficult to be measured from the two original outputs. By comparing Figure 8a,b, we can find that after the original signal is processed by the bistable system stochastic resonance model, the signal characteristic of the signal is significantly enhanced. The two states of the SR-processed signal have an obvious difference and have a large noise tolerance. Moreover, the dynamic part is mainly composed of periodic signals, which correspond to the stage of the sensor sliding over the surface ridge texture. Others parts are the static parts that are caused by the bistable system stochastic resonance model driven by random noise. The signal state carried out transition switching according to the Kramers transition rate *rk*, which depends on the noise distribution and intensity. Therefore, we can easily obtain the time interval through correlation analysis. Comparing Figure 8c,d, it can be found that the stochastic resonance method does not affect the principal frequency of the original periodic signal and makes its amplitude more prominent than other frequencies, which is recognized as noise. The difference between the amplitude of the principal frequency corresponding to the two sensing cells is caused by the difference in the structure during manufacturing.

Combining the information obtained above, we can figure out the spatial period and ridge height information of the texture according to the method proposed in Section 3.1. The calculation result of the space period of each sample is shown in Figure 9a. The average estimation error of the space period is 0.0217 mm, the average relative estimation error is 1.085%, and the maximum estimation error is 0.0465 mm. The largest estimation error occurs in the experiment of the texture sample with the smallest ridge height. The finer the texture, the more seriously the signal generated by the tactile sensor is affected by noise. The SR method plays an auxiliary role in noise reduction. Figure 9b shows the relationship between the principal frequency amplitude of the signal output by sensing cell 3# for each sample and the corresponding ridge height. It is observed that the principal frequency amplitude is positively correlated with the ridge height. The output signal amplitude of the same ridge height experiment fluctuates to a certain extent because there are manufacturing height errors in the silicon wafer sample, and the fluctuation of the static part also affects the result. However, their influence will decrease with the increase in the data volume of the dynamic part. At this end, we can establish the mapping relationship between the ridge height and amplitude to detect the ridge height of the texture.

The ridge height detection experiment of the texture samples is as follows. From the SR-processed signal output by tactile cell 3#, as shown in Figure 8b, five cycles of signals are arbitrarily intercepted and processed by a threshold comparator. The output result is shown in Figure 10a. This result removes the unstable part of the SR-processed signal and preserves the high-level part, which corresponds to the ridge in the texture sample to the greatest extent. The ratio of the high level to the total signal length in the processing result of the output signal for each sample generated by sensing cell 3# is calculated, and the relationship between the result and *n* is shown in Figure 10b. Theoretically, the calculated ratio should be equal to *n*. However, because the SR method has certain randomness, the high-level part of the signal cannot accurately represent the ridge width, and the errors of the calculated ratio are large. However, it can be seen from Figure 10b that when the intervals between the ridge widths are large, there is an obvious mapping relationship between the average ratio and *n*. Thus, the ridge width of the ridge texture sample can be obtained from the output signal.

## 5. Discussion

In the existing research on fine texture detection using tactile technology, HB Muhammad et al. [12] successfully distinguished ridge texture samples with a ridge height of 200 μm and a spatial period as low as 200 μm. Yasutoshi Takekawa et al. [17] succeeded in obtaining the spatial period and ridge height information of ridge texture samples with ridge heights as low as 25 μm. In this article, we tested fine ridge texture samples with different ridge heights (0.9, 4, 10 μm) and different ridge widths (0.3, 0.5, 0.7, 1 mm) but with the same spatial period (2 mm). The mean relative error of the estimation of the space period was 1.085%. The ridge width and ridge height had a mapping relationship with their corresponding parameters, respectively, and the error tolerances are acceptable. Under more systematic experiments, the feature information of the fine texture with a ridge height as low as 0.9 μm can be detected, which indicated that our tactile sensor coupled with the SR method has the potential to achieve the detection of texture with a sub-micron or lower ridge height.

More strikingly, we have completed the detection task of fine textures with fewer data sets based on a sensor array with a simple structure and a simple method. In 2012, Jeremy A. Fishel et al. [19] achieved the classification of 16 natural textures through simple tactile fingers supplemented by Bayesian exploration methods, with an accuracy of 99.6%. In 2017, Udaya B. Rongala et al. [21] used neural network methods to identify 10 natural textures, including glass materials with a grain size of less than 90 nm. The aforementioned methods all require a large number of data sets and a complex learning and training process and neglect the detailed information of the texture, but just judge the texture category of a given sample. The SR method we chose can detect weak periodic signals under the condition of a shorter data set, and the output results have obvious signal characteristics for subsequent calculations. This method generally has an effect on low-frequency signals, but the frequency of periodic signals for texture detection may be as high as 300 Hz. Thus, when the frequency of the periodic signal to be detected is high, the frequency detection range can be increased by performing a normalized scale transformation on the bistable system.

In more detail, the noticeable difference in mind caused by external stimulus difference in human finger touches can be used to evaluate the fine texture detection capability of the sensor. In 2012, Kadir Beceren et al. [25] validated the enhancement effect of SR on human dynamic tactile through a series of psychophysical experiments. The smallest noticeable difference is 1.6 μm under the condition of no noise input but reached 0.93 μm after adding the appropriate noise, which is similar to the detection capability of our tactile sensor after SR enhancement.

## 6. Conclusions and Outlook

In this article, a flexible solid–liquid composite tactile sensor array developed by our laboratory was used to systematically investigate the performance of tactile sensors in fine ridge texture detection through a bistable system stochastic resonance model, including estimation of the spatial period and recognition of ridge height and ridge width. Our experiment used several fine ridge texture samples to provide a weak periodic stimulus to validate the detection capability of the tactile sensor array. The results show that this method can significantly improve the signal characteristic of the output signal. Regarding the detection of texture information, the results show that the mean relative error of the estimation for the spatial period is 1.085%, and the ridge width and ridge height, respectively, have a monotonic mapping relationship with the corresponding model output parameters. Compared with the human finger experiment, the tactile sensor has a similar capability to sensing fine textures, which shows that the sensor designed in the laboratory has a very promising application prospect.

In future work, the following aspect can be improved and expanded on the work of this article: explore the accurate estimation method of ridge width and ridge height, consider the self-adaptability of the model parameters of the stochastic resonance bistable system, and extend the fine texture detection method to non-periodic ridge texture.

## Figures and Tables

**Figure 1 micromachines-13-00440-f001:**
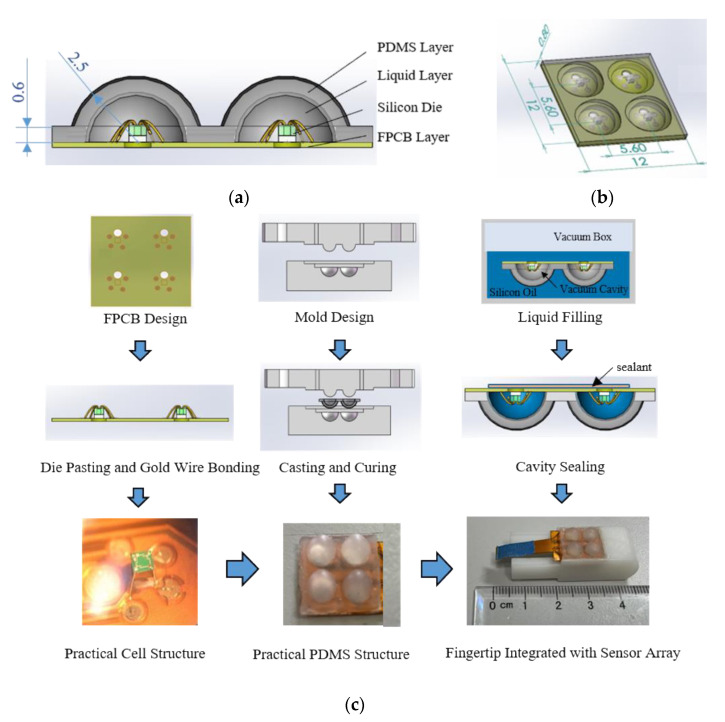
Schematic diagram of sloid–liquid composite tactile sensor array structure: (**a**) cross-section diagram of the sensing cell structure; (**b**) diagram of the whole tactile sensor array; (**c**) flow diagram of sensor fabrication process.

**Figure 2 micromachines-13-00440-f002:**
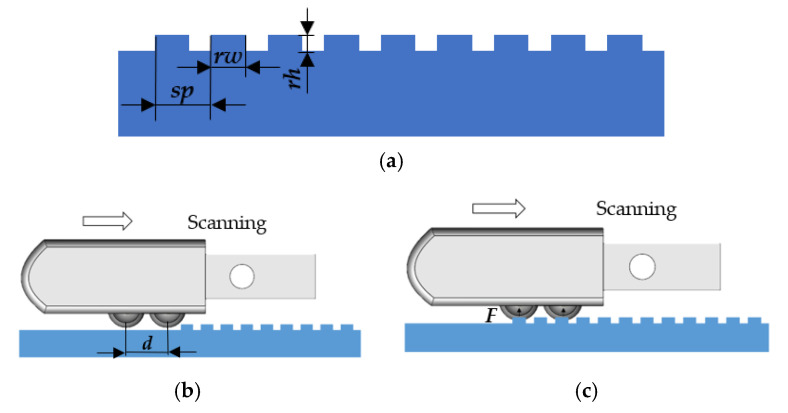
Schematic diagram of texture detection of tactile sensor array: (**a**) cross-section diagram of the ridge texture sample; (**b**) fingertip integrated with tactile sensor array starting to slide; (**c**) ridge of texture compressing the sensing cell.

**Figure 3 micromachines-13-00440-f003:**
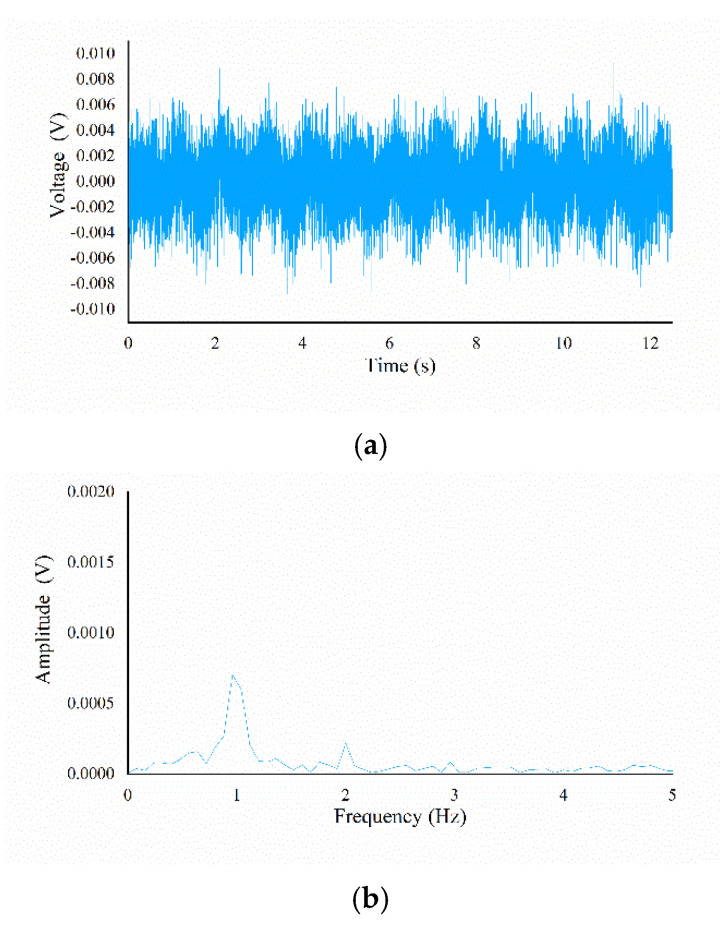
Diagram of tactile signal containing texture information: (**a**) time-domain signal of scanning result; (**b**) frequency-domain signal of scanning result.

**Figure 4 micromachines-13-00440-f004:**
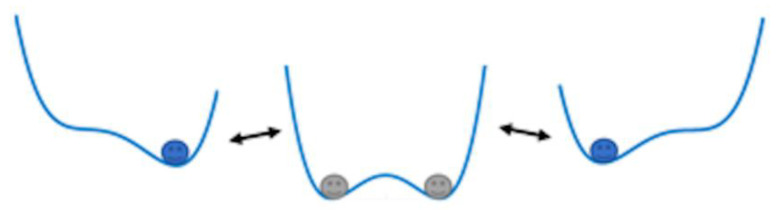
Diagram of stochastic resonance effect of bistable system.

**Figure 5 micromachines-13-00440-f005:**
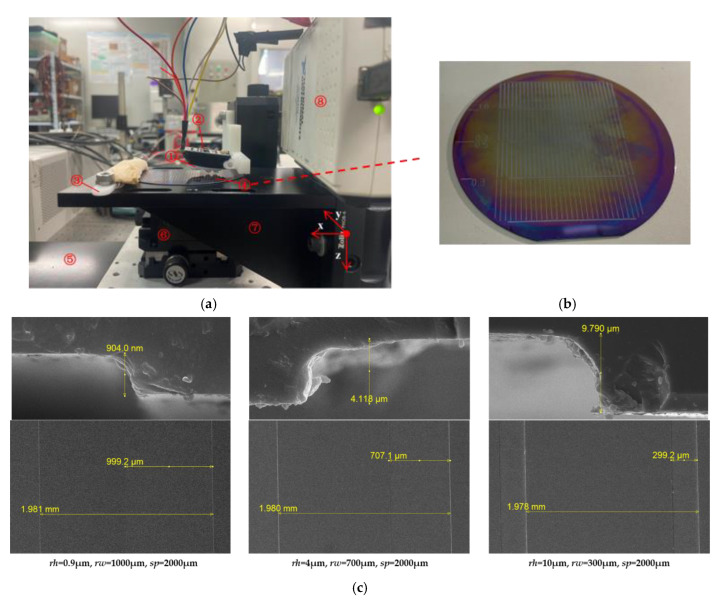
Experimental loading platform for texture detection: (**a**) ① prosthetic fingertip integrated with tactile sensor array, ② conditioning circuit, ③ NI-DAQ device, ④ texture sample, ⑤ fixed stage, ⑥ three-axis precision linear stage, ⑦ long-travel linear motor stage, ⑧ clamp; (**b**) silicon wafer with ridge texture; (**c**) SEM image of the micro-structure of the silicon wafer.

**Figure 6 micromachines-13-00440-f006:**
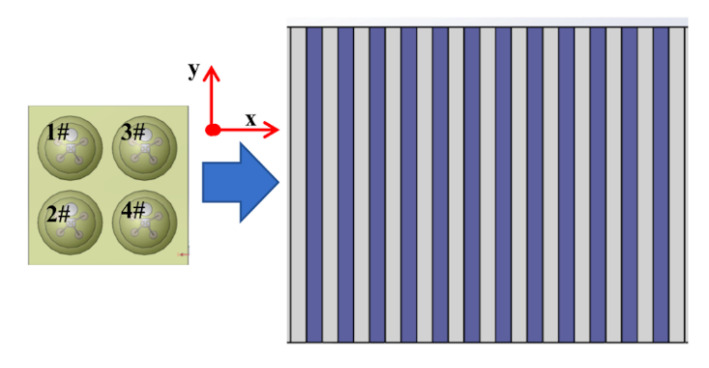
Tactile sensor sliding across the ridge texture sample.

**Figure 7 micromachines-13-00440-f007:**
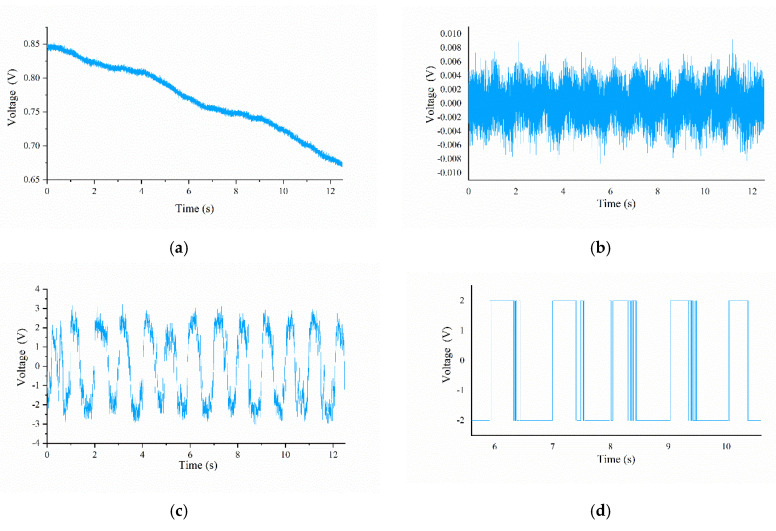
Diagram of tactile signal processing flow: (**a**) diagram of row output signal; (**b**) original signal processed by de-baseline method; (**c**) signal processed by SR model; (**d**) signal segment passed through software-designed threshold comparator.

**Figure 8 micromachines-13-00440-f008:**
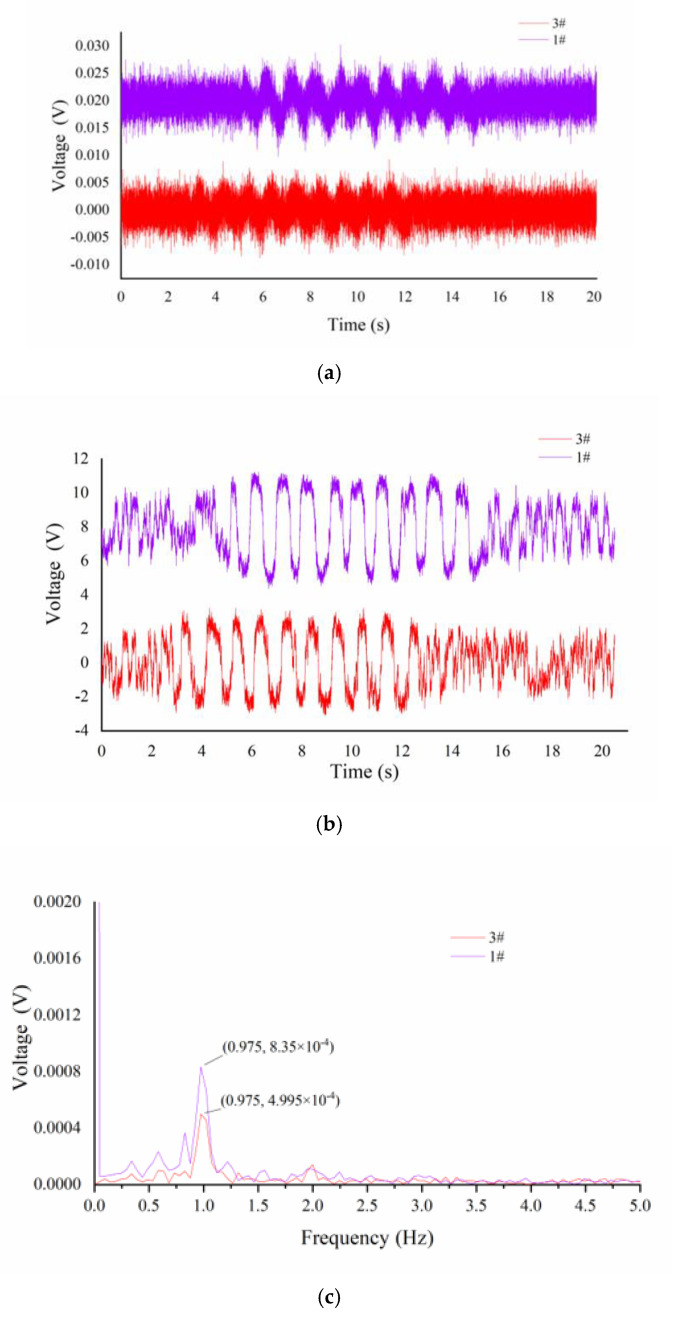
Diagram of output signal for sample S31 (v = 2 mm/s): (**a**) original signal processed by de-baseline method; (**b**) signal processed by SR model; (**c**) original frequency-domain signal; (**d**) SR-processed frequency-domain signal.

**Figure 9 micromachines-13-00440-f009:**
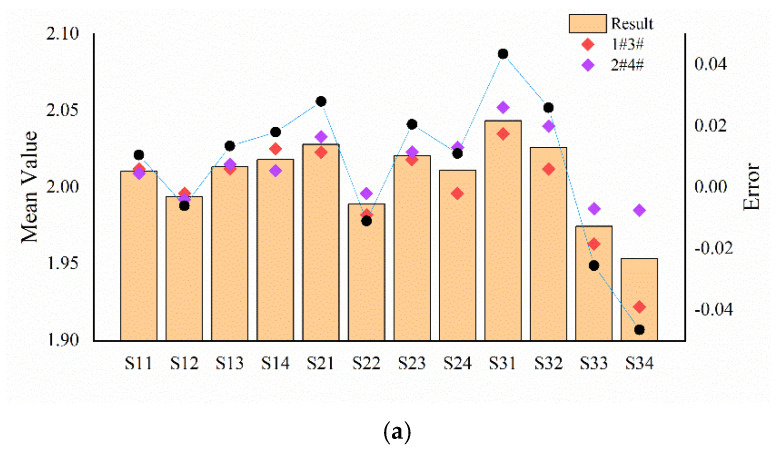
Experimental result of texture detection: (**a**) the result of spatial period estimation; (**b**) the result of ridge height recognition.

**Figure 10 micromachines-13-00440-f010:**
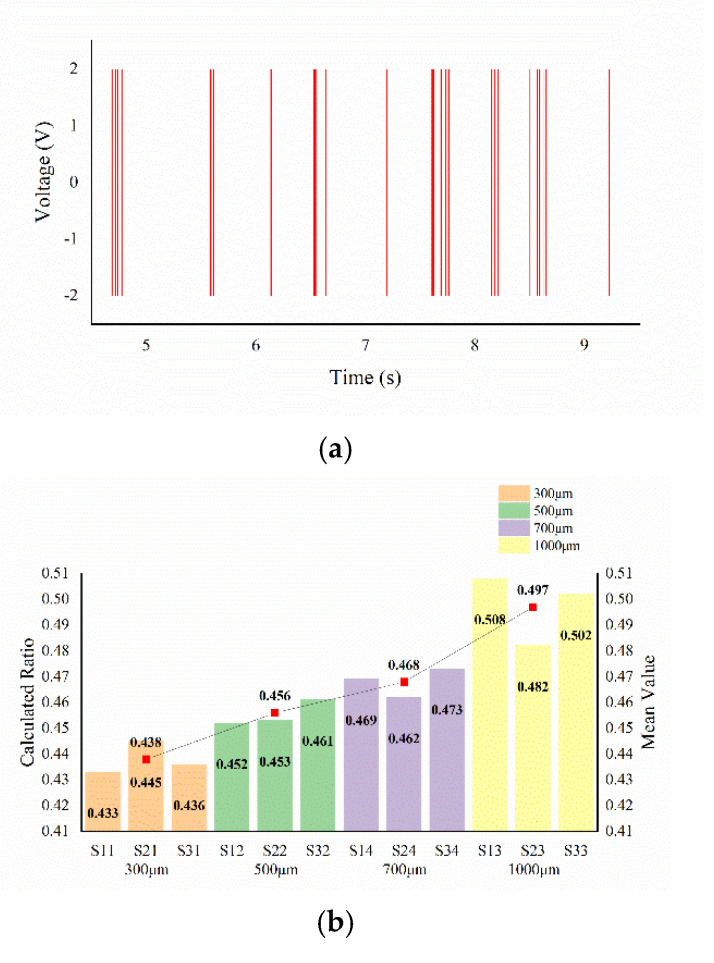
Experimental result of texture sample ridge width detection: (**a**) signal segment passed through threshold comparator; (**b**) result of ridge width recognition.

**Table 1 micromachines-13-00440-t001:** Parameters of ridge texture sample.

NUMBER	RIDGE HEIGHT *RH* (μM)	RIDGE WIDTH *RW* (μM)	SPATIAL PERIOD *SP* (μM)
**S11**	10	300	2000
**S12**		500	
**S13**		700	
**S14**		1000	
**S21**	4	300	
**S22**		500	
**S23**		700	
**S24**		1000	
**S31**	0.9	300	
**S32**		500	
**S33**		700	
**S34**		1000	

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
