# Peer review of "Fine Texture Detection Based on a Solid–Liquid Composite Flexible Tactile Sensor Array"

_micromachines, 2022, doi:10.3390/mi13030440_

Round 1

Reviewer 1 Report

an extensive rephrasing may improve readability: for instance, it's not recommended to start sentences with " And"  after a fullstop. sometimes there is lacking of the verb. a few baroque sentences with repeated concepts are here. an example is "and the detection ability of the tactile sensor array is limited by the sensitivity of the sensor array (lines 193-194)." this is absolutely useless. makes only more stressing reading the paper. conciseneess is a key value.

also, it's not so clear which are the merits of your device: is it cheap? easily scalable? seems a little bulky, what about installing it on a prostethic device? do you foresee improvements with wider arrays?

why you choose h<10microns? (also, fomatting of variables with "inverted commas" is awful, as well as two-letters variable name, and DeltaP is quite misleading since it recalls more a difference in pitch rather than a pitch and is not recalling a spatial frequency)

lines 86-92 sounds a little bit confusing, needs rephrasing

replace "the autors"  with "we"

clarify what do you mean with attenuation effect and which is the relationship with sensitivity

I'd suggest clean up the first part of tactile sensor description at lines 120-131, which can be almost entirely skipped with a few observations integrated in the following part. the sensor fabrication part is to be edited to make it readable, which is not the case.

figure 7 must be in one page

what about the detectability of ridges with smaller pitches? this seems not being addressed here, putting only 2 mm pitch.

from figure 8c and 8d it is not apparent the benefit of the SR approach in terms of the S/N ratio, thus statement n line 393 is wrong. which peaks are related to the surface features and which are just noise? there is even no comparison between your algorythm and the conventional FFT (i.e. some thing like fig. 9 for both approaches)

also, in fig. 9 it is quite unclear the correlation between the error and the ridge heigth.

Author Response

We greatly appreciate for your valuable comments. The answers to the questions and comments are given in the word file. Please see the attachment.

Reviewer 2 Report

The paper reported the detection of the geometrical features of micro structures using piezoresistive tactile sensor. 

1) Overall, the current mauscript is like a student laborotary report with too many details but missing the key points. The texts should be rewritten and reorganized, and try to be as concise as possible. At the beginning of each paragraph, provide a topic sentence, so that it is easier for the reader to follow the idea. Extensive editing of English language is required.

2) The texts in some figures are too small, and the terminology of the parameters such as height, width, amplitude should be consistent in texts and in all the figures.

3) Your results are lacking of validation. Please provide SEM images of the micro structures and measure the geometrical features e.g. width, height, spacing and compare with your detection results. Without validation, the credibility of the results are questionable.

Author Response

(The authors gave the same response as above.)

Reviewer 3 Report

The manuscript "Fine Texture Detection Based on a Solid-liquid Composite Flexible Tactile Sensor Array" fabricated a solid-liquid composite flexible tactile sensor to detect surface texture. A stochastic resonance algorithm was applied to improve the signal-to-noise ratio and the three ridge parameters were discussed. It is very interesting and meaningful work. However, the language should be polished and some statements were overrated. The specific comments are listed as the following:

  • The authors claimed that this method can be used to detect sub-micron fine ridge texture. However, only the four specimens (S31-S34) with ridge height of 0.9um were discussed. No specific discussions were made based on these specimens. Only an overall relative error for the spatial period was estimated for all the specimens. Based on this information, it is difficult to conclude that current method is able to detect the texture in sub-micron level.
  • Although different heights and widths were designed to investigate the detectability, only some trends were observed. It is not sufficient enough to conclude that this research is able to obtain the texture information, especially the height and width.
  • For the spatial period, the results of S31 and S34 show relative larger error. The author claims that “The smaller the texture height, the greater the estimation error of the sample”. But why the errors of S32 and S33 are smaller?
  • Line 220, x (spatial) is the variable for V(x) in equation (2), while t (time) is the variable for s(t). Please check whether something is wrong. Besides, how to determine the parameters (a=b=16, K=130000) in V?
  • Line 275, “After inputting the original signal and the Gaussian white noise with a certain noise intensity into the bistable system stochastic resonance model”, how to generate the noise signal? Is it the same for all the signal?
  • Line 339, “It is observed that the principal frequency amplitude is positively correlated with the ridge height of the texture”. The authors should discuss Figure 9(b) with more details. The amplitude seems decrease with the ridge height, but the quantitative change should be discussed and the variation in the same height should be mentioned.
  • Line 350, the name of y-axis is missing. Moreover, it seems different software are applied to get the figures (a) and (b). Not only the font but also the numeric are not consistent. The authors should check thoroughly the whole manuscript.
  • There are lots of Language errors in the whole manuscript. Such as “good in frequency response” in Line 128; “After the signal is conditioned by the conditioning circuit” in Line 140; “fuse … and …” in Line 200; “a timely manner” in Line 201; “According to…, determined…” in Line 224; “the just noticeable difference which reflects the noticeable difference” in Line 379.

Author Response

(The authors gave the same response as above.)

Round 2

Reviewer 1 Report

many parts of the description are still to be imprved. it's a pity since the method described is interesting. it's quite tiring to read and interpret every sentence. please have a mother tongue revise your paper.

Reviewer 2 Report

English language and spell need to be checked carefully.

Reviewer 3 Report

The previous comments were well addressed. But further editing of English language and style is required.